# ARROW: Allele-Specific Recombined gRNA Design for Reduced Off-Target with Enhanced Specificity

**DOI:** 10.3390/bioengineering12111237

**Published:** 2025-11-12

**Authors:** Taegeun Bae, Kyung Wook Been, Seunghun Kang, Sumin Hong, Junho K. Hur, Woochang Hwang

**Affiliations:** 1Department of Genetics, College of Medicine, Hanyang University, Seoul 04763, Republic of Korea; btg417@naver.com (T.B.); ysbkw@hanyang.ac.kr (K.W.B.); 2Hanyang Biomedical Research Institute, Hanyang University, Seoul 04763, Republic of Korea; 3Graduate School of Biomedical Science and Engineering, Hanyang University, Seoul 04763, Republic of Korea; rkdekffyd@naver.com (S.K.); sumin1716@naver.com (S.H.); 4Hanyang Institute of Bioscience and Biotechnology, Hanyang University, Seoul 04763, Republic of Korea; 5Department of Pre-Medicine, College of Medicine, Hanyang University, Seoul 04763, Republic of Korea

**Keywords:** CRISPR, allele-specific genome editing, cancer-associated point mutation

## Abstract

Background/Objectives: Allele-specific genome editing using the CRISPR–Cas9 system is crucial for achieving precise therapeutic interventions in dominant inherited diseases that are otherwise difficult to treat with conventional approaches. However, Cas9–guide RNA (gRNA) complexes often tolerate single-base mismatches in target sequences, making it challenging to discriminate between wild-type and mutant alleles differing by only one nucleotide. Although previous studies have attempted to improve specificity by introducing mismatches into gRNAs, none has systematically investigated the impact of different mismatch types and positions on editing outcomes. In this study, we developed an effective strategy to enhance specificity and minimize off-target effects by deliberately introducing mismatches into gRNAs and comprehensively evaluating their editing performance. Results: We established an efficient strategy for the selective editing of mutant alleles that reduces Cas9 sequence tolerance and enhances specificity through the intentional introduction of mismatches into gRNAs. The efficacy of this approach was demonstrated by successful allele-specific editing of cancer-associated heterozygous point mutations in EGFR L858R and KRAS G12V, while minimizing editing of the corresponding wild-type alleles. Conclusion: Compared with perfectly matched gRNAs, the strategic incorporation of mismatches into gRNAs enhanced editing specificity for single-base mutant alleles. Our findings substantially improve the precision and safety of CRISPR-based genome editing for cancer therapy, particularly in cases involving mutant alleles.

## 1. Introduction

The Clustered Regularly Interspaced Short Palindromic Repeats (CRISPR) system, originally an adaptive immune mechanism in bacteria and archaea that protects against bacteriophage infection, has been developed into a powerful and versatile genome-editing platform [1]. Among various CRISPR-associated nucleases, the *Streptococcus pyogenes* Cas9 (SpCas9) protein exhibits outstanding genome-editing efficiency in mammalian cells [2,3]. The CRISPR–Cas9 system has been widely applied to the treatment of genetic diseases caused by pathogenic DNA mutations [4,5,6].

Typically, a gRNA perfectly complementary to the on-target sequence is employed to achieve efficient genome editing while minimizing off-target effects. However, the specificity of perfectly matched gRNAs is often lower than expected, as the Cas9–gRNA complex can tolerate single-nucleotide mismatches and fail to accurately distinguish between alleles differing by only one base pair [7,8,9,10,11,12,13,14,15,16,17,18]. This mismatch tolerance complicates allele-specific targeting, particularly in heterozygous genetic disorders.

To address this limitation, several approaches have been proposed to enhance precision in CRISPR–Cas9–based editing by exploiting the sequence-dependent nature of Cas9 activity [19,20,21,22,23,24,25,26]. For instance, Koo et al. designed a gRNA that specifically recognized the EGFR L858R mutation in a mouse model [25], and Smith et al. utilized allele-specific gRNAs in induced pluripotent stem cells (iPSCs) to selectively correct the JAK2-V617F mutation [26]. Nonetheless, a single-nucleotide difference between wild-type and mutant alleles often remains insufficient to overcome the inherent sequence tolerance of Cas9.

In this study, we propose a novel strategy that enhances allele discrimination by deliberately introducing intentional mismatches into gRNAs targeting cancer-associated point mutations. This approach, termed ARROW (Allele-specific Recombined gRNA design for Reduced Off-target With enhanced specificity), is designed to reduce Cas9 tolerance and enable precise mutant allele editing while avoiding cleavage of the corresponding wild-type allele. By incorporating one or two intentional mismatches into gRNAs perfectly matched to the mutant target, ARROW aims to achieve higher editing specificity with minimal off-target activity.

Although similar concepts have been explored previously [24,25,26], no study has systematically analyzed how different mismatch types and positions affect Cas9 editing outcomes. Here, we evaluated single-mismatch gRNAs designed for the EGFR L858R and KRAS G12V mutations. Our results demonstrate that mismatched gRNAs were highly efficient at editing mutant alleles while exhibiting reduced activity on wild-type alleles. Furthermore, we confirmed that intentional mismatch introduction effectively minimized off-target effects without significantly compromising on-target efficiency.

Collectively, our findings establish ARROW as an effective and generalizable design strategy for allele-specific genome editing, providing a valuable framework for enhancing the precision and safety of CRISPR-based gene therapy.

## 2. Materials and Methods

### 2.1. Cell Subculture and Transfection of Dual Fluorescence Reporter Vector System

HEK293T and SW403 cells were cultured in DMEM media (Welgene, Gyeongsan si, Republic of Korea) supplemented with 10% fetal bovine serum (FBS) and 1% antibiotics. H1975 cells were maintained in RPMI 1640 media (Welgene) containing 10% FBS and 1% antibiotics. For cell transfection, 1.5 × 10^5^ cells were seeded in 24-well plates and transfected with 200 ng of SpCas9 plasmid (Addgene #104171), 200 ng of gRNA plasmid (Addgene #104174), and 100 ng of the reporter vector using Lipofectamine 2000 (Invitrogen, Waltham, MA, USA). Cells were cultured for 48 h before FACS analysis for mRFP and eGFP expression.

### 2.2. Generation of Mismatched gRNAs

Normally, a gRNA (i.e., an original gRNA) is designed to perfectly match the target DNA to correctly edit the target sequence, not the other sequences in the genome. Our gRNAs were optimized for improved single-nucleotide discrimination between wild-type and mutant alleles by introducing mismatches into the original gRNA. There are three possible mismatch bases at each locus in a 20 bp long gRNA, resulting in all 60 possible one-base mismatched gRNAs of on-target sequences. The design and generation of 60 potential mismatched gRNAs for each target would be labor-intensive, so we introduced mismatches into the seed or non-seed regions of CRISPR-Cas9. The mismatches at 3 positions in the seed region (−1, −3, −6 bp from PAM) and 3 non-seed positions (−9, −12, −15 bp from PAM) were randomly introduced with 4 types of bases (A, T, C, G).

### 2.3. Targeted Deep Sequencing

All sequencing experiments were performed in duplicate. PCR amplified on- and off-target sites using Phusion Hot Start II polymerase (NEB, Ipswich, MA, USA). The primary PCR products were used for a second PCR with NGS adapter primers, followed by a third PCR with indexing primers. PCR amplicons were purified and sequenced by Illumina paired-end sequencing. Sequencing data was analyzed using Cas-Analyzer (http://www.rgenome.net). Indel frequencies near the PAM site (NGG) were considered CRISPR-Cas9-induced mutations. For sequence analyses, insertions and deletions located 3 bp upstream of the PAM were considered mutated by CRISPR-Cas9. All sequencing results are compared between groups using un-paired *t*-test in GraphPad PRISM 8. Statistical significance is denoted with asterisks (*, *p* < 0.05) in the figures.

### 2.4. T7E1 Assay for Potential Off-Target Sites

Cas-OFFinder (http://www.rgenome.net) was used to find potential off-target sites that differed from on-target sequences by up to 8 nt and that differed by up to 2 nt with a 1- to 5-nt DNA or RNA bulge. PCR amplicons were obtained for the predicted off-target candidates from genomic DNA extracted after the treatment of cells with CRISPR-Cas9. For each amplicon, the denaturation and re-annealing processes were carried out using a PCR machine, followed by the DNA cleavage reaction with the T7E1 enzyme at 37 °C for 25 min. The reaction was terminated by adding a stop buffer (100 mM EDTA, 1.2% SDS). The cleaved DNA fragment was separated by 2% agarose gel electrophoresis.

## 3. Results

### 3.1. Characterization of the Insertion and Deletion Rates of gRNA Variants on EGFR L858R and KRAS G12V Mutations

Our aim was to develop an allele-specific CRISPR targeting strategy for pathogenic DNA sequences that contains cancer-related EGFR L858R and KRAS G12V mutations. To identify the most suitable gRNA candidates for our study, it is necessary to observe the pattern of overall editing efficiency caused by mismatches on various positions within the gRNAs. As described in the methods section (Figure 1A), we used a surrogate dual fluorescence reporter vector system. The designed reporter vectors were then transfected into HEK293T cells with gRNAs containing different mismatches, and the editing frequency was then measured by targeted deep sequencing (Figure 1B).

The mutated base of EGFR L858R is located in position 12 and KRAS G12V is located in position 1 from PAM of each gRNA. We assessed the editing frequencies of perfectly matched and mismatched gRNAs for wild-type and mutant alleles of EGFR and KRAS mutations through the reporter system. In general, the dual fluorescence reporter vector assays revealed higher insertion and deletion (indel) rates for perfectly matched gRNAs to target sequences than for those with a mismatch. However, the indel rates for mismatched gRNAs are still quite high: 1–10% for EGFR and 40–60% for KRAS (Appendix A). Similarly, gRNAs that were perfectly matched to mutant alleles exhibited higher indel rates on mutant sequences than on wild-type sequences. However, despite a single base difference, these gRNAs still cleaved wild-type sequences of EGFR and KRAS, albeit to a lesser extent (Appendix A).

First, we observed the editing efficiency of the gRNAs that were perfectly matched to the mutant sequences when targeting the wild-type sequences (Figure 2A–D). The gRNA that is perfectly matched to the wild-type sequence of EGFR (5′-UUUUGGGCUGGCCAAACUGC-3′) exhibited an indel rate of 40% on the wild-type sequence. The gRNA mismatched to the wild-type sequence by a single base, carrying the L858R mutant nucleotide and perfectly matching the L858R mutant sequence (5′-UUUUGGGCGGGCCAAACUGC-3′), still induced an indel rate of 11% (Figure 2A,B). This indicates that the gRNA perfectly matched to the EGFR L858R mutation cannot discriminate the single-base difference between the wild-type and mutant alleles. Similarly, the gRNA perfectly matched to the wild-type KRAS sequence (5′-CUUGUGGUAGUUGGAGCUGG-3′) exhibited indel rates of 77% on the wild-type sequence. The gRNA containing the KRAS G12V mutated sequence (5′-CUUGUGGUAGUUGGAGCUGU-3′), which differs from the wild-type sequence by a single base, exhibited an indel rate of 66% (Figure 2C,D). In other words, gRNAs containing either the wild-type or mutant sequences still targeted the wild-type alleles of each gene. These findings indicate that the specificity of CRISPR genome editing using perfectly matched gRNA is insufficient for single-nucleotide differences of EGFR L858R and KRAS G12V mutations.

### 3.2. Intentionally Mismatched gRNA Can Discriminate Between Mutant and Wild-Type Alleles

Leveraging insights from the editing frequencies of mismatch gRNAs, we sought to develop gRNAs optimized for improved single-nucleotide discrimination between wild-type and mutant alleles. We tested a strategy involving the introduction of an additional mismatch into the gRNA, with the goal of achieving allele-specific genome editing and greater single-nucleotide selectivity. In the assay, we sought to determine whether addition of deliberate mismatches in the PAM-proximal seed region might provide improved specificities compared to introducing mismatches in the PAM-distal region. To this end, we investigated the effects of different mismatch types, considering variations in both nucleotide identity and loci, between the target DNA and gRNAs. Intentional mismatches were introduced into the gRNAs at six positions, located 1, 3, 6, 9, 11 (or 12), and 15 bp from the PAM site (Figure 3A). We then evaluated the targeting efficiency to determine which mismatches between the target DNA and gRNAs have a greater effect on the indel rates.

As a result, the highest indel rates were observed for most of targets when using the perfectly matched gRNA, as indicated by the “X” marks in the heat maps. (Figure 3B,C). Most of mismatches resulted in reduced indel rates, regardless of the nucleotide type or their positions within the gRNAs. As exceptions, position-15 and -3 mismatched gRNA targeting the KRAS wild-type sequence exhibited higher indel rates compared with the perfectly matched gRNAs (Figure 3C). The results demonstrated that the editing efficiency of SpCas9-mediated genome editing is influenced by both mismatch position and type. In addition, although introducing an additional mismatch into gRNAs affected the targeting efficiency of each gene, the resulting changes in indel rates were insufficient to achieve allele-specific genome editing capable of discriminating single-base differences.

gRNAs were categorized based on their changes in indel rates and assessed the indel rate changes to identify suitable gRNAs for each target: at each position, the mismatch type with the smallest reduction in indel rate was defined as Highly tolerant Mismatch (H), whereas the mismatch causing the largest reduction was defined as Low tolerant Mismatch (L). To enhance the ability to distinguish single-base differences in target sequences, we designed mismatched gRNAs by introducing selected H and L mismatches at positions 1, 3, 6, 9, 11, or 15 bp from the PAM site into the perfectly matched gRNA sequences targeting the EGFR L858R and KRAS G12V mutations (Figure 4A,B). Accordingly, the mismatched gRNAs were constructed to have one-base mismatches with the mutant allele and two-base mismatches with the wild-type allele for each gene. The designed gRNAs were expected to edit the mutant sequences but fail to edit the wild-type sequences due to the presence of two mismatches between the gRNAs and the wild-type targets. In other words, this strategy was expected to enhance single-nucleotide discrimination between the wild-type and mutant sequences.

Introduction of a single mismatch into the perfectly matched gRNA targeting EGFR L858R mutant sequence decreased the indel frequency on the mutant allele. However, the gRNAs containing H showed higher indel frequency than those containing L, regardless of the mismatch position within the gRNA (Figure 4A). The mismatched gRNA containing an H at position-9 showed the highest indel frequency (~45%) when compared to the other mismatched gRNAs; however, it also induced editing of the wild-type allele (~4%). The wild-type allele was not edited at any position except with the P9-H gRNA. Based on these results, we selected four two-base mismatch gRNAs (P3-H, P6-L, P9-H, and P15-H), each carrying two mismatches relative to the EGFR wild-type sequence. These gRNAs exhibited high specificity while maintaining indel frequency on the mutant target (Figure 4A) and will be used to validate whether intentional mismatches can enhance gRNA specificity in cell lines.

The specificity of the H- and L-mismatched gRNAs targeting the KRAS G12V mutation was also evaluated. Five positions within the KRAS G12V–targeting gRNA were selected, and the indel frequencies of the two-base mismatch gRNAs against the KRAS wild-type sequence were analyzed. As observed for EGFR L858R (Figure 4B), the mismatched gRNAs containing H exhibited higher indel frequencies than those containing L. However, unlike the EGFR L858R–targeting gRNAs, which barely edited the wild-type allele, the mismatched gRNAs targeting the KRAS G12V mutation were prone to editing the wild-type allele despite the introduction of mismatches. The findings demonstrated that the tolerance of two-base mismatched gRNAs to the wild-type sequence varied according to position, mismatch type, and target gene. Accordingly, four gRNAs (P3-H, P6-L, P6-H, and P9-H) were chosen for their high specificity and maintained indel frequencies and will be tested in subsequent experiments (Figure 4B).

### 3.3. Intentionally Mismatched gRNAs Enhance Specificity in Mammalian Cells

To test our strategy, we used four cell lines: H1975 cells carrying the EGFR L858R mutant allele, SW403 cells with the KRAS G12V mutant allele, and HEK293T cells lacking these mutations. We evaluated whether the selected one-base mismatched gRNAs targeting mutant sequences of each gene could specifically edit the mutant alleles in mammalian cells.

We transfected the selected the mismatched gRNAs and SpCas9 expression plasmid into HEK293T cells and H1975 cells carrying EGFR L858R mutation. We then measured the editing frequency using targeted deep sequencing. In HEK293T cells, high editing efficiency was observed when the perfectly matched gRNA targeting the wild-type sequence was used (indel rate ~23%). In contrast, minimal editing occurred when the perfectly matched gRNA for the mutant sequence was used (indel rate ~1.5%), consistent with the absence of these mutations in HEK293T cells. As expected, no editing event occurred with two-base mismatched gRNAs on the wild-type EGFR allele (Figure 5A). However, in H1975 cells carrying the EGFR L858R mutation, several mismatched gRNAs exhibited indel frequencies comparable to those of the perfectly matched gRNA for the mutant sequence. Among them, the P9-H mismatched gRNA demonstrated the highest discrimination ability, inducing indel rates of ~8% on the EGFR L858R allele while showing only ~0.28% on the wild-type EGFR allele (Figure 5B).

Next, selected mismatched gRNAs for KRAS G12V and SpCas9 cassette plasmids were then transfected into HEK293T cells and SW403 cells. Similarly, the one-base mismatched gRNAs against the mutant sequence barely edited HEK293T cells containing only wild-type alleles (Figure 5C). However, the perfectly matched gRNA for the mutant sequence induced a detectable editing level (~12%) in HEK293T cells, indicating weak allele selectivity between the wild-type and mutant sequences. In SW403 cells, the single-base mismatched gRNAs targeting the mutant allele (P3-H and P6-H) specifically induced indels at rates of ~4–5%, while no indels were detected in HEK293T cells (Figure 5D). Collectively, these results indicate that, although the introduction of intentional mismatches into perfectly matched gRNAs targeting the mutant allele reduced overall editing efficiency, this strategy enhances specificity and is therefore optimal for selectively targeting the mutant allele.

### 3.4. Intentionally Mismatched KRAS G12V gRNAs for Decreasing Potential Off-Target Effects

We anticipated that the mismatched gRNAs generated using our design strategy might also increase the number of mismatches to some of the potential off-target sites. To evaluate whether introducing intentional mismatches into gRNAs decreases off-target activity, we performed in vitro cleavage assays using PCR-amplified fragments spanning the predicted off-target sites for both perfectly matched and mismatched gRNAs. The off-target effects of perfectly matched and mismatched gRNAs targeting the KRAS mutant allele across multiple predicted off-target sequences (Figure 6A–D).

We conducted in vitro cleavage assays on eight off-target PCR fragments of the KRAS G12V mutant target using a perfectly matched gRNA and a P6-H mismatched gRNA (Figure 6A–D). The perfectly matched gRNA (Mut gRNA) cleaved the wild-type and mutant templates as well as five off-targets (1, 2, 4, 7, and 8) without discrimination (Figure 6A). In contrast, P6-H gRNA preferentially cleaved the mutant template while showing markedly reduced cleavage rate on the wild-type and off-target templates (Figure 6B,C). Collectively, these results indicate that introducing intentional mismatches into the KRAS G12V gRNAs decreased the off-target effect for some loci, likely by increasing the number of mismatches between the gRNA and the target DNA sequences.

## 4. Discussion

Several studies have shown that the Cas9-gRNA complex has difficulties in discriminating one-base differences due to its high tolerance [7,8,9,10]. This remains one of the major challenges complicating precise medical genome editing using CRISPR-Cas9. To address this issue, a trial-and-error approach has often been employed to achieve specific editing of target genes. A commonly explored strategy involves altering gRNA sensitivity through chemical modifications; however, this method is costly and impractical for widespread use. Furthermore, the potential adverse effects of such chemical modifications in vivo have not been thoroughly evaluated.

To overcome the sensitivity limitations of sequence-dependent approaches, we developed a strategy to enhance specificity by introducing intentional mismatches into gRNAs complementary to the target DNA. This approach offers high stability, as the sensitivity can be finely tuned without the need for chemical modification of gRNAs. Moreover, unlike chemical modifications, which are costly and time-consuming to manufacture, intentional mismatches can be introduced easily and efficiently. We demonstrated that introducing intentional mismatches into gRNAs significantly improves the specificity of targeting heterozygous mutations, including EGFR L858R and KRAS G12V, which are otherwise difficult to distinguish from wild-type sequences due to single-base differences.

Our finding is in line with the study by Fu et al., which reported that differences in mismatch tolerance among targets is due to the change in free energy between the target DNA and the Cas9-gRNA complex [10]. The study further suggested that this effect is influenced by the GC content of the target sequence, as GC pairs have higher bond energies compared to AT pairs. Consistent with these findings, our data show that the EGFR gene, which displays lower tolerance, is comparatively AT-rich. This effect is particularly evident in the PAM-distal region, where base mismatches are more likely to destabilize the Cas9–gRNA–target DNA complex.

Finally, we demonstrated that introducing intentional mismatches into gRNAs has the possibility to reduce off-target effects, resulting in potentially increased specificities. In Figure 6, the in vitro cleavage assays of KRAS G12V targets showed that gRNAs carrying intentional mismatch resulted in reduced cleavage of the off-target PCR templates compared to the perfectly matched gRNAs. Yet, we anticipate that the observed reduction in the KRAS G12V off-target activity may be context-dependent and could vary with the sequence composition and similarity of the on-target and off-target sites.

In summary, we anticipate that the allele-specific genome editing strategy, termed ARROW (Allele-specific Recombined gRNA design for Reduced Off-target With enhanced specificity via intentional mismatches), is a tool that can improve the safety and precision of CRISPR-Cas9-based genome editing.

## Figures and Tables

**Figure 1 bioengineering-12-01237-f001:**
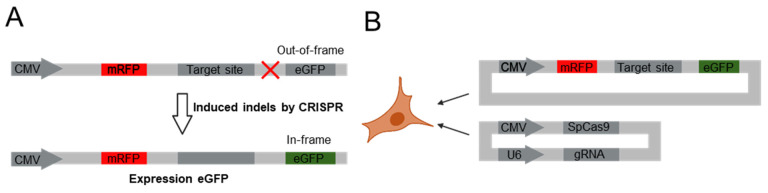
Schematic representation of dual fluorescence reporter vector system. (**A**) The dual fluorescence reporter vector has out-of-frame codon between target site and eGFP sequence, resulting in expression of mRFP alone. When CRISPR induces indels on target site, the out-of-frame is changed to in-frame codon and eGFP is expressed with mRFP. (**B**) In all cell experiments, cells are transfected by dual fluorescence reporter vector, SpCas9- and gRNA-expressing vectors.

**Figure 2 bioengineering-12-01237-f002:**
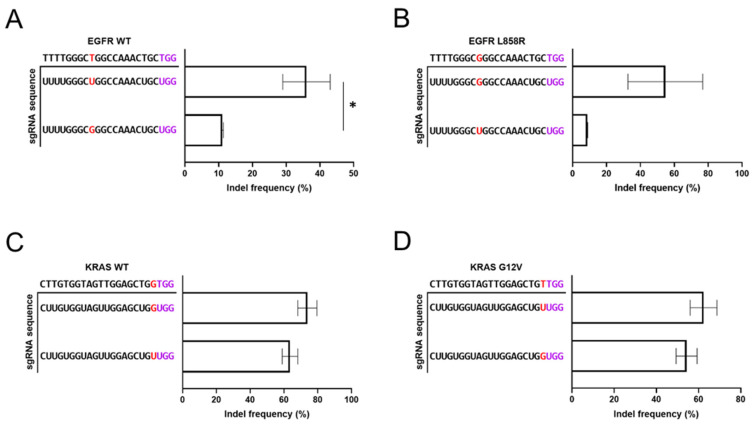
The editing efficiency of gRNAs perfectly matched to wild-type or mutant sequences of cancer-associated genes. The graphs indicate the editing efficiency of gRNAs harboring either the wild-type or mutant sequences of (**A**,**B**) EGFR and (**C**,**D**) KRAS genes. These results show that a single-nucleotide difference is not sufficient to discriminate between the wild-type and mutant sequences of cancer-associated genes. Statistical significance is denoted with asterisks (*, *p* < 0.05).

**Figure 3 bioengineering-12-01237-f003:**
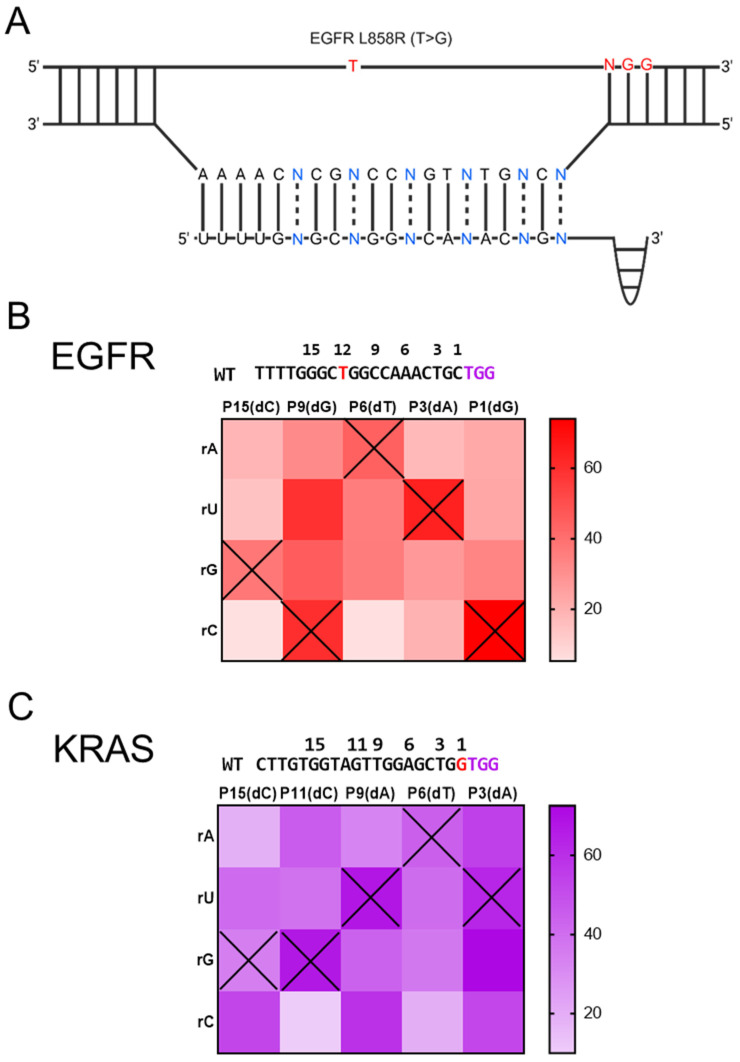
Optimization of CRISPR activity using mismatched gRNA targeting wild-type EGFR and KRAS genes. (**A**) Schematic representation of the EGFR L858R mutation site (T to G) indicated as example for design of mismatched gRNA. (**B**,**C**) Heat-maps depict editing efficiency of gRNAs containing mismatches at various positions and nucleotides (dN = DNA base of mismatch position; rA, rU, rG, rC = RNA bases). Higher editing efficiency is represented by darker shades and “X” marks are pointed to perfectly matched gRNA on target sequence.

**Figure 4 bioengineering-12-01237-f004:**
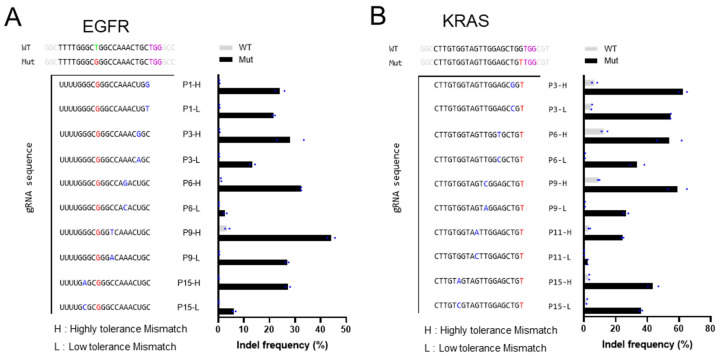
Comparison of indel rates of mismatched gRNAs targeting EGFR and KRAS mutations. (**A**,**B**) For all target genes, one intentional mismatch is introduced into the gRNAs at six positions, located 1, 3, 6, 9, 11 (or 12), and 15 bp from the PAM site. Accordingly, the mismatched gRNAs were constructed to have one-base mismatch with the mutant allele and two-base mismatches with the wild-type allele for each gene. At each position, the mismatch type with the smallest reduction in indel rate was defined as H (Highly tolerant Mismatch), whereas the mismatch causing the largest reduction was defined as L (Low tolerant Mismatch).

**Figure 5 bioengineering-12-01237-f005:**
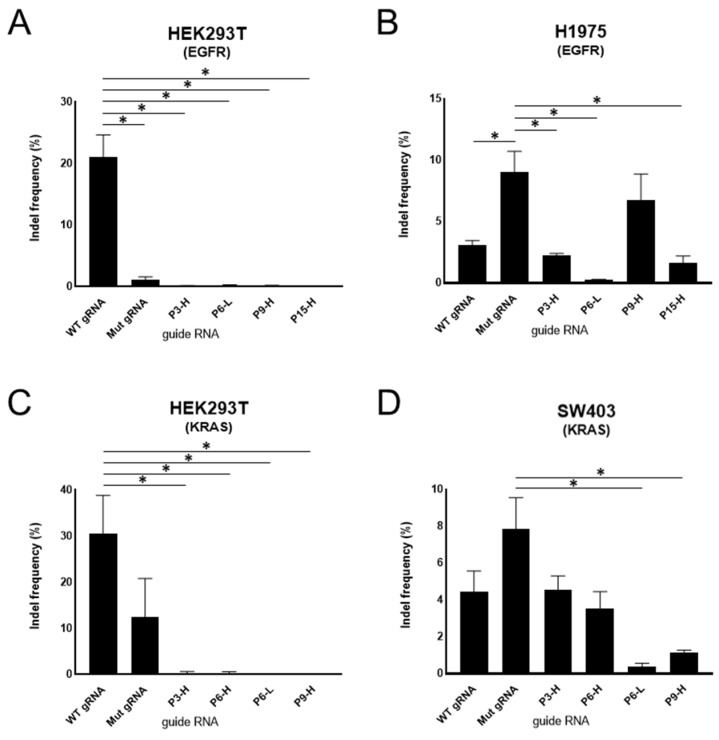
Allele specific editing efficiency of mismatched gRNAs in mammalian cell lines. The graphs show the indel rates observed in mammalian cell lines when SpCas9 and gRNAs are delivered. (**A**,**C**) HEK293T cells contain no mutant allele of EGFR and KRAS genes. (**B**) H1975 cells carry the EGFR L858R mutant allele and (**D**) SW403 cells harbor the KRAS G12V mutant allele. Wild-type (WT) gRNA means gRNA perfectly matched to wild-type sequence and Mutation (Mut) gRNA means gRNA perfectly matched to mutant sequence of each gene. The others indicate one-base mismatched gRNAs against Mut gRNA. The one-base mismatched gRNAs distinguished wild-type sequence and seldom induced indels compared to WT gRNAs. Among the one-base mismatched gRNAs, P9-H gRNA of EGFR and P3-H, P6-H gRNAs of KRAS induced indels as much as Mut gRNAs. Statistical significance is denoted with asterisks (*, *p* < 0.05).

**Figure 6 bioengineering-12-01237-f006:**
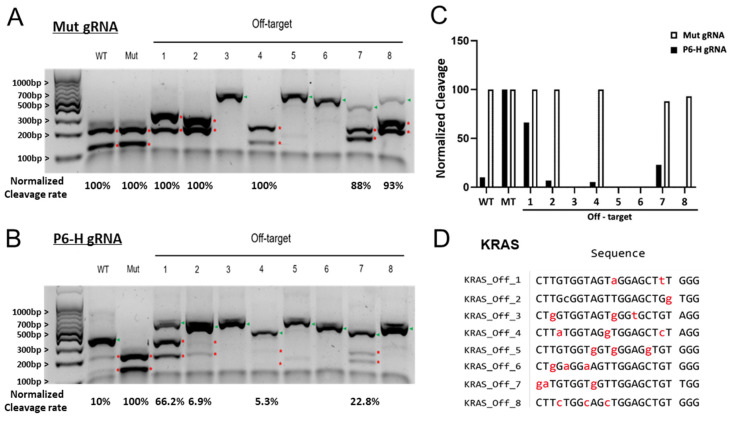
Reduced off-target effects of mismatched gRNAs targeting KRAS mutation. (**A**,**B**) In vitro cleavage of KRAS G12V mutant, KRAS wild-type and eight potential off-target DNA with a gRNA perfectly matched to KRAS G12V mutant allele (Mut gRNA) and mismatched P6-H gRNA. The green arrowheads and red stars indicate uncleaved PCR products and cleavage fragments, respectively. (**C**) Quantification of the ratios of cleaved products of on-target and off-target DNA by mut and P6-H gRNA. (**D**) The sequences of the off-target sequences; mismatches relative to the gRNA are highlighted in red.

## Data Availability

The data presented in this study are openly available in BioProject: PRJNA1335927. [BioProject] [https://www.ncbi.nlm.nih.gov/bioproject/?term=PRJNA1335927 (accessed on 29-September-2025)] [PRJNA1335927].

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
