# Peer review of "ARROW: Allele-Specific Recombined gRNA Design for Reduced Off-Target with Enhanced Specificity"

_bioengineering, 2025, doi:10.3390/bioengineering12111237_

Round 1

Reviewer 1 Report

Comments and Suggestions for Authors

The manuscript entitled “ARROW: Allele-specific Recombind gRNA design for Reduced Off-target With enhanced specificity” reports an optimized CRISPR-Cas9 gRNA design strategy (ARROW) aiming to reduce off-target and increase specificity. The topic is important and valuable for gene therapy. However, the overall presentation are not yet sufficient and need to be major revisied. I recommend major revision.

  1. The ARROW strategy is interesting but not entirely novel; similar “intentional mismatch” approaches were already proposed in earlier allele-specific CRISPR works (e.g., Smith et al., 2019; Koo et al., 2020). The authors should clearly distinguish what is conceptually new in ARROW compared to existing mismatch-based gRNA optimization methods.
  2. There is a lack of experimental validation demonstrating that ARROW outperforms previous design strategies. Only EGFR and KRAS examples are shown, with limited quantitative comparison. Please provide additional examples or statistical validation.
  3. The description of gRNA design and screening methods is too brief. What criteria were used for mismatch positions and types? How were on/off-target efficiencies measured? Detailed methodology (number of replicates, controls, detection limits) should be included to ensure reproducibility.
  4. The claim that ARROW reduces off-target effect is not convincingly supported. Please provide experimental off-target mapping (e.g., GUIDE-seq, Digenome-seq, or computational prediction with validation). Otherwise, the conclusion remains too speculative.
  5. The abstract and introduction include several redundant sentences and awkward phrasing (for example, lines 16–20 and 53–56). Please rephrase to improve readability and flow.
  6. Some acronyms appear before definition (e.g., gRNA, iPSCs). The manuscript would benefit from language polishing by a native or professional editor.

Reviewer 2 Report

Comments and Suggestions for Authors

    The authors present a  study on the development of the ARROW strategy, a novel gRNA design approach for achieving allele-specific genome editing with enhanced specificity and reduced off-target effects. The work is innovative, methodologically sound, and addresses a significant challenge in the therapeutic application of CRISPR-Cas9. The manuscript is generally clear. However, several aspects require clarification and minor improvements before publication.

  1. Materials and Methods section 2.1: Please provide detailed information about the Cas9 plasmid, gRNA plasmid, and reporter plasmid, such as whether they have Addgene numbers.
  2. Section 2.2.4: There is a missing closing parenthesis: "The reaction was terminated by adding a stop buffer (100mM EDTA, 1.2% SDS."
  3. The Materials and Methods section should specify the statistical analysis methods used in the study.
  4. While introducing mismatches reduces off-target effects, does it also impact the on-target editing efficiency?
  5. Are the effects of mismatch position and base type universally applicable, or are they context-specific?
  6. The resolution of the figures needs to be improved.
  7. Line 338: There are two consecutive periods. Please remove one.

Reviewer 3 Report

Comments and Suggestions for Authors

This study provided an interesting method of intentional introduction of mismatch to selectively edit mutant alleles and reduce the editing in wild-type alleles and off-target effect. It may improve the design of gRNA for genomic editing. The overall design and result are fine. Some detail information need to be provided.

  1. “2.3. Targeted deep sequencing”. It is not clear how to calculate Indel frequencies based on the sequencing result of PCR product. Please give clear and detail explanation.
  2. How was the indel rates calculated? The transfection efficiency of SpCas9 plasmid, gRNA plasmid, and the reporter vector was included?
  3. Figure 6. What are the green and red arrow heads? The sizes of cut and uncut product should be illustrated in figure legend.
  4. How did the position of second mutated nucleotide affect the off-target effect? What is the best mutant site for the design of gRNA with minimal off-target effect?

Round 2

Reviewer 3 Report

Comments and Suggestions for Authors

Authors have improved the manuscript.
